# The role of solar and geomagnetic activity in endothelial activation and inflammation in the NAS cohort

Jessica E. Schiff[1]*, Carolina L. Z. Vieira[1], Eric Garshick[2,3,4], Veronica Wang[1], Annelise Blomberg[1], Diane R. Gold[5,6], Joel Schwartz[1], Samantha M. Tracy[1], Pantel Vokonas[7], Petros Koutrakis[1]

1 Department of Environmental Health, Harvard T.H. Chan School of Public Heath, Boston, MA, United States of America, 2 Pulmonary, Allergy, Sleep, and Critical Care Medicine Section, VA Boston Healthcare System, Boston, MA, United States of America, 3 Channing Division of Network Medicine, Brigham and Women's Hospital, Boston, MA, United States of America, 4 Harvard Medical School, Boston, MA, United States of America, 5 Professor of Medicine, Harvard Medical School, Boston, MA, United States of America, 6 Professor of Environmental Health, Harvard T.H. Chan School of Public Health, Boston, MA, United States of America, 7 VA Normative Aging Study, Veterans Affairs Boston Healthcare System and the Department of Medicine, Boston University School of Medicine, Boston, MA, United States of America

* jschiff@hsph.harvard.edu

**Data Availability Statement:** Data cannot be shared publicly because of patient privacy and confidentiality. Access to research data and biospecimens is by request to the principal

## Abstract

This study investigated the associations between solar and geomagnetic activity and circulating biomarkers of systemic inflammation and endothelial activation in the Normative Aging Study (NAS) cohort. Mixed effects models with moving day averages from day 0 to day 28 were used to study the associations between solar activity (sunspot number (SSN), interplanetary magnetic field (IMF)), geomagnetic activity (planetary K index ($K_p$ index), and various inflammatory and endothelial markers. Biomarkers included intracellular adhesion molecule-1 (sICAM-1), vascular cell adhesion molecule-1 (sVCAM-1), C-reactive protein (CRP), and fibrinogen. After adjusting for demographic and meteorological variables, we observed significant positive associations between sICAM-1 and sVCAM-1 concentrations and solar and geomagnetic activity parameters: IMF, SSN, and $K_p$. Additionally, a negative association was observed between fibrinogen and $K_p$ index and a positive association was observed for CRP and SSN. These results demonstrate that solar and geomagnetic activity might be upregulating endothelial activation and inflammation.

## Introduction

Solar activity, which induces disturbances of the Earth's magnetic field, periodically oscillates in cycles of approximately 11-years following patterns of minimum to maximum and maximum to minimum sunspot activity [1, 2]. Exposure to broad-spectrum electromagnetic solar activity can adversely affect human physiology and lead to adverse health outcomes. Recent epidemiological studies have shown solar and geomagnetic activity is associated with chronic diseases, including cardiovascular diseases(CVD) [3, 4]. A meta-analysis of 263 U.S. cities

investigators and is considered on a case-by-case basis from the VA Cooperative Studies Program (CSP). Contact Dr. Pantel Vokonas (Pantel.Vokonas@va.gov) or Dr. Avron Spiro (Avron.Spiro@va.gov) for details. Gross β activity data had been previously collected from the Environmental Protection Agency's (EPA) RadNet program monitors at this website: https://enviro.epa.gov/enviro/erams_query_v2.simple_query The solar data were obtained from spacecraft measurements and can be found on the NASA Goddard Space Flight Center's OMNIWeb at this website: https://omniweb.gsfc.nasa.gov/form/dx1.html.

**Funding:** P.K. United States Environmental Protection Agency grant RD-835872 https://www.epa.gov/grants National Institutes of Health grant R21 ES029637 https://www.nih.gov/grants-funding The funders had no role in study design, data collection and analysis, decision to publish, or preparation of the manuscript.

**Competing interests:** The authors have declared that no competing interests exist.

demonstrated that short-term geomagnetic disturbances are significantly associated with deaths from cardiovascular disease and myocardial infarction [2]. Another study showed significant correlations between space weather and hypertensive mortality and demonstrated cyclicity in the association [2, 5]. Astronauts exposed to the high energy particles from cosmic rays have demonstrated vascular endothelial dysfunction independent of weightlessness and also have an increased CVD mortality rate approximately 4–5 times higher than astronauts who did not travel into space and remained in low Earth's orbit under the protection of the atmosphere [6]. While these studies suggest an association between solar variables and cardiovascular outcomes, the impact of solar activity on biomarkers for CVD, including biomarkers for endothelial function and inflammation in a large cohort has not been previously studied.

It is known that various radiation sources, including space radiation, can influence cardiovascular risk, however there is an absence of research on the effects of solar and geomagnetic activity on endothelial function and inflammatory markers, and the role it could play in the development of atherosclerosis and cardiovascular disease [7]. Therefore, we explored the association between short- and middle-term exposures to solar activity parameters and biomarkers for adverse endothelial function and inflammatory responses. This study used soluble intracellular adhesion molecule-1 (sICAM-1) and vascular adhesion molecule-1 (sVCAM-1) as biomarkers of endothelial activation, C-reactive protein (CRP) as a marker of inflammation, and fibrinogen as a marker of coagulation. Each of these biomarkers has been associated with increased risk of cardiovascular outcomes [8–12]. The solar activity measures included interplanetary magnetic field (IMF) intensity and sunspot number (SSN), while planetary K index ($K_p$) was used as an indicator of geomagnetic disturbances. This study complements recent research evaluating the associations between solar and geomagnetic activity and immune function and is the first study to our knowledge that specifically investigates the associations between solar activity and endothelial and inflammation [13]. While inflammation and immune response are physiologically closely related and changes in white blood parameters could impact inflammatory and endothelial parameters and vice versa, this study focuses only on the impacts of solar and geomagnetic activity on endothelial parameters.

## Methods

### Study population

Study subjects were selected from the Normative Aging Study (NAS), a closed, longitudinal cohort of elderly men established in 1963 in the Greater Boston Area by the U.S. Department of Veterans Affairs [14, 15]. At recruitment, men were free of chronic disease at baseline, had a mean age of 42 (range 21–81), and had subsequent examinations every 3 to 5 years. The participants abstained from smoking, were asked to eat a fat-free meal the night before, and completed an overnight fast prior to their clinical examination [16]. Examinations occurred on Tuesdays and Wednesdays and the dates that the bloods were collected were matched to the dates of the solar and geomagnetic activity, air pollution, and weather measurements. A complete physical exam and laboratory testing, including phlebotomy, were conducted at each clinical visit. Standardized questionnaires were administered to each participant to collect data about medical history, smoking history, and alcohol consumption. Only participants with recorded observations between May 2000 and December 2017 were included in the study. Furthermore, study participants with a CRP >10 mg/L were excluded from analysis, since CRP levels >10 mg/L could be indicative of an acute infection or other illness [17]. The final study population included 742 participants with 2,273 observations. The study received approval from both Institutional Review Boards of the Harvard T.H. Chan School of Public Health and

the VA Boston Healthcare System and all participants signed an informed consent form prior to study enrollment.

## Blood measurements

Biomarkers evaluated in this study included sICAM-1, sVCAM-1, CRP, and fibrinogen. sICAM-1 and sVCAM-1 are used as biomarkers of endothelial activation. They are surface receptors on endothelial cells that bind white blood cells and lymphocytes to the endothelium, generating inflammation and plaque buildup [18–22]. Elevated sICAM-1 has been associated with increased risk of myocardial infarction and coronary death and significantly elevated levels of both sICAM-1 and sVCAM-1 have been found in patients with slow coronary flow, suggesting that endothelial activation and inflammation contribute to adverse cardiac outcomes [9, 12]. CRP is a well-studied marker of inflammation and infection and is often used as a marker for CVD and other inflammatory diseases [23–26]. Increased CRP levels have been associated with increased risk for symptomatic peripheral arterial disease and elevated risk of coronary heart disease mortality [8, 11]. Fibrinogen, a marker of coagulation, is pro-inflammatory and plays a key role in the clotting cascade, contributing to the development of atherosclerotic plaques [27, 28]. Fibrinogen plays an important role in platelet aggregation and is a risk factor for cardiovascular disease with a meta-analysis showing that the risk of CVD events was twice as high in those with the highest tertile of plasma fibrinogen concentrations compared to those in the lowest tertile [10].

Plasma sICAM-1 and sVCAM-1 concentrations (ng/L) were measured using an enzyme-linked immunoabsortbent assay method (R&D Systems, Minneapolis, MN) [29]. High sensitivity CRP concentration (mg/L) was measured using an immunoturbidimetric assay on a Hitachi 917 analyzer using reagents and calibrators from Roche (Roche Diagnostics—Indianapolis, IN) with a limit of detection of 0.03 mg/L [29]. Fibrinogen concentration (mg/dL) was measured using a thrombin reagent, MDA Fibriquick. Biomarker concentrations were normalized using log transformations prior to analysis as distributions were skewed.

## Solar and geomagnetic activity data

The solar and geomagnetic activity exposures examined in this study included Interplanetary Magnetic Field (IMF), sunspot number (SSN), and the geomagnetic parameter, $K_p$ index. IMF is a measure of the portion of the Sun's magnetic field that is carried into interplanetary space due to solar wind and it has a 27 day periodicity, matching the length of time for a single solar rotation [30]. The total strength of the IMF is defined as the combined measurements of IMF strength from all directions; north-south, east-west, and towards-sun and away-from sun. When there are periods of increased solar activity there is an increased number of sunspots, larger total IMF values, shock waves in the interplanetary medium, solar energetic particle events (e.g., solar flares, coronal ejections, etc.), and geomagnetic disturbances [31, 32]. Sunspots are defined as temporary dark areas in the Sun's photosphere where solar magnetic field concentrations interact with the Sun's plasma, with the number of sunspots correlating to solar activity level [1, 33]. The number and size of sunspots describes the 11-year solar cycle periods [1]. $K_p$ index was used to characterize global geomagnetic activity in Earth's magnetic field, quantifying disturbances in the Earth's magnetic field.

Measurements for the $K_p$ index are made every 3 hours using ground-based magnetometers and are measured on a 0 to 9 scale, with 0 being interpreted as very little geomagnetic activity and 9 as extreme geomagnetic activity [34]. The $K_p$ value that is reported is the maximum fluctuation, which is the maximum positive and negative deviations of geomagnetic activity that occur during the 3-hr window [34]. These solar and geomagnetic data were obtained from spacecraft measurements and can be found on the NASA Goddard Space Flight Center's OMNIWeb [35].

## Air pollution data

We examined how the impact of solar and geomagnetic activity on the biomarkers were influenced by exposure to $PM_{2.5}$ (particulate matter with diameter $\leq 2.5$ μm, in μg/m$^3$), black carbon (BC, in μg/m$^3$), and particle number concentrations (PN, #/cm$^3$) over Boston during the study period. We specifically limited the pollution analysis to particulate matter constituents as prior research suggests that solar activity may interact with particulate matter, and thus we adjusted for these pollutants in our analysis [36]. Daily concentrations of these air pollutants were measured at the Harvard supersite located on the roof of the Countway Library of Medicine at Harvard Medical School. This site is approximately 3 miles from downtown Boston and is located approximately 1 kilometer from the Hospital where the NAS cohort examinations took place. Daily $PM_{2.5}$ samples were collected by the Harvard Impactor Sampler [29, 37]. Black carbon concentrations were continuously measured using an aethalometer (Magee Scientific Corp, Model AE-21, Berkeley CA). Particle number (PN) concentrations were continuously measured using a particle condensation particle counter (CPC, TSI Inc. Model 3022a, Shoreview, MN).

## Particle radioactivity

Gross β activity (mBq/m$^3$) is a measure of all ambient particle bound β–emitting radionuclides [38, 39]. Data were collected from the Environmental Protection Agency's (EPA) RadNet program monitors [29, 39]. The RadNET program includes 140 monitors across all 50 U.S. states that contain a total suspended particle high-volume air sampler collecting particle bound β–emitting radionuclides [29, 39]. In this study, we used log transformed β activity from the Boston, MA, RadNet site to control for the potential effects of particle radioactivity and assessed moving averages of 0 to 28 days.

## Weather assessment

Models were adjusted for daily ambient temperature (˚C) and relative humidity (%RH). Local, daily weather data from Boston Logan International Airport Monitors were collected from the National Oceanic Atmospheric Administration's (NOAA) National Climatic Data Center. Adjustment of weather variables was limited to temperature and humidity because those variables are the most common meteorological parameters evaluated in similar studies evaluating the impacts of solar and geomagnetic activity on health outcomes [2, 13, 40]. Furthermore, the authors did not have access to other parameters, such as air pressure.

## Solar and geomagnetic activity exposure windows

We calculated moving averages from day 0 (day of visit) to 28 days to allow for evaluation of the short-term effects of solar activity and biomarker concentration because moving averages allowed for control of a lag in the exposure-outcome relationships. Additionally, prior environmental exposure studies (radiation and air pollution) have demonstrated that health outcomes from exposures do not always appear at the time of exposure, and can manifest in the following days [41–43].

## Statistical analysis

In this study, we used mixed effects linear models to allow for random intercepts for each subject to investigate the associations between solar and geomagnetic activity and biomarker concentrations. All biomarkers were log transformed to normalize the distribution. The general

statistical model can be written as:

$$Y_{ij} = \beta_0 + b_{0i} + \beta_1 X_{ij} + \beta_2 X_{2ij} + \beta_3 X_{3ij} + \ldots + \beta_p X_{pij} + \varepsilon_{ij}$$

where $Y_{ij}$ is the logarithm of the biomarker concentration for subject i at measurement j, $\beta_0$ is the primary intercept, $b_{0i}$ is random intercept for each subject, $\beta_1$ is the main effect of solar radiation, and $\beta_2 \ldots \beta_p$ are the effects of measured covariates $X_{2ij}$ through $X_{pij}$. $\varepsilon_{ij}$ is the error term.

To assess the estimated response in endothelial activation and inflammatory biomarkers per interquartile range (IQR) increase in solar activity variable, the final mixed effects linear models included the following variables: age (yr), body mass index (BMI) (kg/m$^2$), day of the year (assigned 1–365 or 1–366), day of the week, temperature (˚C), relative humidity (%RH), seasonality (was controlled with the model variable: Sin(2πDOY/365) + Cos(2πDOY/365), race category (white or non-white), hypertension diagnosis (yes or no), statin medication use (yes or no), lung conditions (yes or no), diabetes diagnosis (yes or no), history of chronic heart disease (yes or no), history of stroke (yes or no), smoking category (never, current, or former), and two or more drinks per day (yes or no). Models were adjusted according to previous NAS analyses examining air pollution effects on the same biomarkers [44]. The effect estimates were presented as percent change from baseline in endothelial activation and inflammatory outcome per IQR increase in solar variables. A sensitivity analysis was conducted to evaluate if moving from Massachusetts impacted the results as some participants moved from Massachusetts to other states after the cohort was established, although most returned to Massachusetts for the follow up visits.

Secondary analyses were performed to estimate the effects of air pollution and β -radiation (PM$_{2.5}$, BC, PN, or log β) in our models. The most significant air pollutant and log β moving averages were analyzed in the absence of solar and geomagnetic activity parameters to determine the most significant ambient air pollutants for each biomarker. This pollutant was then used in the secondary statistical models with solar variables (shown in S1 Table). The effect estimates for the associations between biomarkers and solar activity, both without and with air pollution variables, were examined for 0 to 28 day moving averages. Residuals were examined to examine for deviations from linearity. All analyses were performed using R software 3.6.3.

## Results

### Descriptive results

The study population consisted of 742 individual subjects with 2,273 visits between May 2000 and December 2017. The sensitivity analysis of location of residence revealed slightly larger effect estimates when restricted to only those residing in Massachusetts, however, the number of observations in the study decrease to 1,653 (compared to 2,273), reducing our power. We elected to include all participants because the changes in effect estimates were minimal and to preserve power.

The mean age at baseline was 73.3 years (SD = 6.8) and the mean BMI was 28.2 kg/m$^2$ (SD = 4.0). At baseline, the study population was 98.1% white. Of the 742 individuals, 54 (7.3%) had a history of stroke, 108 had diabetes (14.6%), 536 (72.2%) had hypertension, 285 (38.4%) were on statin medication, 223 (30.1%) had coronary heart disease (CHD), 140 (18.9%) consumed two or more alcoholic drinks per day, 34 (4.6%) were current smokers, and 136 (18.4%) had a lung condition.

At first visit, mean sICAM-1 concentration was 311.5 ng/L (SD = 80.8), mean s-VCAM-1 was 1,085.1 ng/L (SD = 377.8), mean CRP was 2.3 mg/L (SD = 2.0), and mean fibrinogen 342.7 mg/dL (SD = 78.2). The mean ambient temperature was 12.8˚C (SD = 8.5), and the mean

**Table 1. Descriptive characteristics of NAS cohort and the weather, 2000–2017.**

|  | First Visit* (n) | All Visits (n) |
|---|---|---|
| Total Unique Examinations | 742 | 2,273 |
| Total Unique IDs | 742 | 742 |
|  | Mean ± SD | Mean ± SD |
| Age (y) | 73.3±6.8 | 76.7±7.0 |
| BMI (kg/m$^2$) | 28.2±4.0 | 27.9±4.1 |
| Relative humidity (%) | 69.1±17.2 | 68.0±17.4 |
| Ambient Temperature (○C) | 12.8±8.5 | 13.2±8.8 |
| sICAM-1 (ng/L) | 311.5±80.8 | 281.9±78.5 |
| sVCAM-1 (ng/L) | 1,085.1±377.8 | 1,042.8±401.3 |
| CRP (mg/L) | 2.3±2.0 | 2.0±1.9 |
| Fibrinogen (mg/dL) | 342.7±78.2 | 334.7±73.2 |
|  | n (%) | n (%) |
| Race |  |  |
| White | 722 (98.1) | 2,206 (97.9) |
| Non-White | 14 (1.9) | 47(2.1) |
| Stroke | 54 (7.3) | 189 (8.3) |
| Diabetes | 108 (14.6) | 375 (16.5) |
| Hypertension | 536 (72.2) | 1,766 (77.7) |
| CHD | 223 (30.1) | 815 (35.9) |
| Statin Medication | 285 (38.4) | 1,221 (53.7) |
| Alcohol Consumption (≥2 drinks daily) | 140 (18.9) | 407 (17.9) |
| Smoking status |  |  |
| Current | 34 (4.6) | 90 (4.0) |
| Former | 476 (64.3) | 1,465(64.6) |
| Never | 230 (31.1) | 714 (31.5) |
| Chronic Lung Diseases | 136 (18.4) | 399 (19.6) |

* Baseline visit; first visit in the study period

relative humidity was 69.1% (SD = 17.2) on the day of data collection. Descriptive statistics on participant demographics are shown in Table 1.

Descriptive statistics of the solar activity measures (IMF and SSN), GMD ($K_p$ index), air pollutants ($PM_{2.5}$, BC, PN) and β radiation are shown in S2 Table. The mean IMF was 5.9 nT (SD = 2.9), the mean $K_p$ index was 18.9 nT (SD = 12.1), and the mean SSN were 81.5 (SD = 71.3). The mean $PM_{2.5}$ concentration was 9.4 μg/m$^3$ (SD = 6.4), the mean BC concentration was 0.74 μg/m$^3$ (SD = 0.40), and the mean PN concentration was 20,963 #/cm$^3$ (SD = 11,852), and the mean log β activity was -5.0 (SD = 0.3).

## Associations of solar and geomagnetic activity and endothelial dysfunction and inflammatory biomarkers

### Intracellular adhesion molecule-1 (sICAM-1)

Overall, IQR increases in SSN, IMF, and $K_p$ were associated with a significant, positive percent change in sICAM-1 concentration (Fig 1A–1C). On the 28th-day moving average, an IQR increase in SSN was associated with a 0.569% (0.360, 0.779) change of sICAM-1 concentration; an IQR increase in IMF was associated with a 10.1% (8.34,11.9) change of sICAM-1 concentration; and an IQR increase in $K_p$ was associated with a 0.414% (0.342, 0.486) percent change of

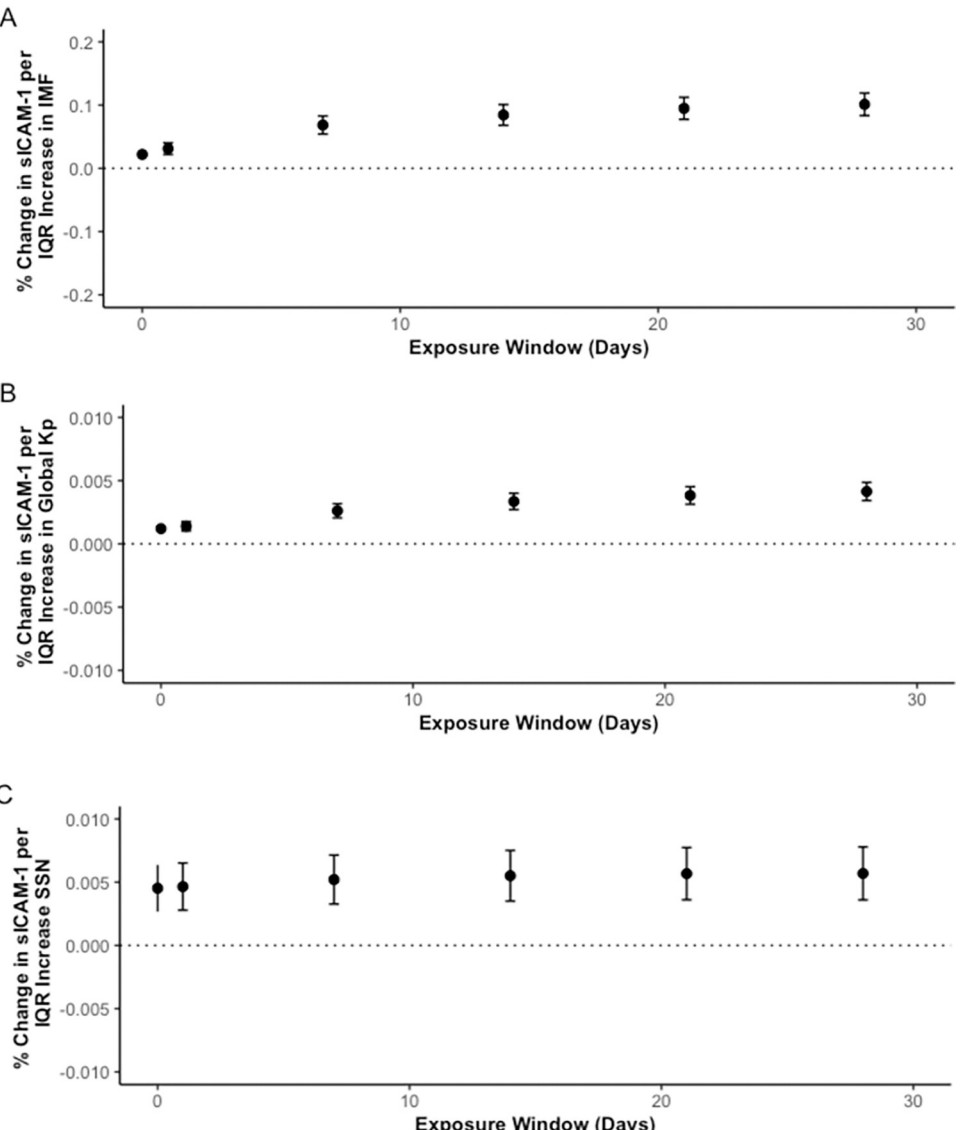

**Fig 1.** Percent change in sICAM-1 concentrations per IQR increase in IMF (A), $K_p$ index (B), and SSN (C) over 28 day moving averages. * y-axes have different scales.

sICAM-1 concentration. Results were of similar magnitude and direction when air pollutants were included in the model. To see all results, refer to S3 Table.

## Vascular cell adhesion molecule-1 (VCAM-1)

Similar to sICAM-1, IQR increases in SSN, IMF, and $K_p$ index were associated with a significant percent change of sVCAM-1 concentration (Fig 2A–2C). On the 28th moving day average an IQR increase in SSN was associated with a 1.45% (1.14,1.75) percent change of sVCAM-1 concentration; an increase in IQR of IMF was associated with a 15.3% (12.7, 17.8) change of sVCAM-1 concentration; and an IQR increase in $K_p$ index was associated with a 0.569% (0.463,0.676) change of sVCAM-1 concentration. Results were of similar magnitude and direction when air pollutants were included in the model. To see all results, refer to S4 Table.

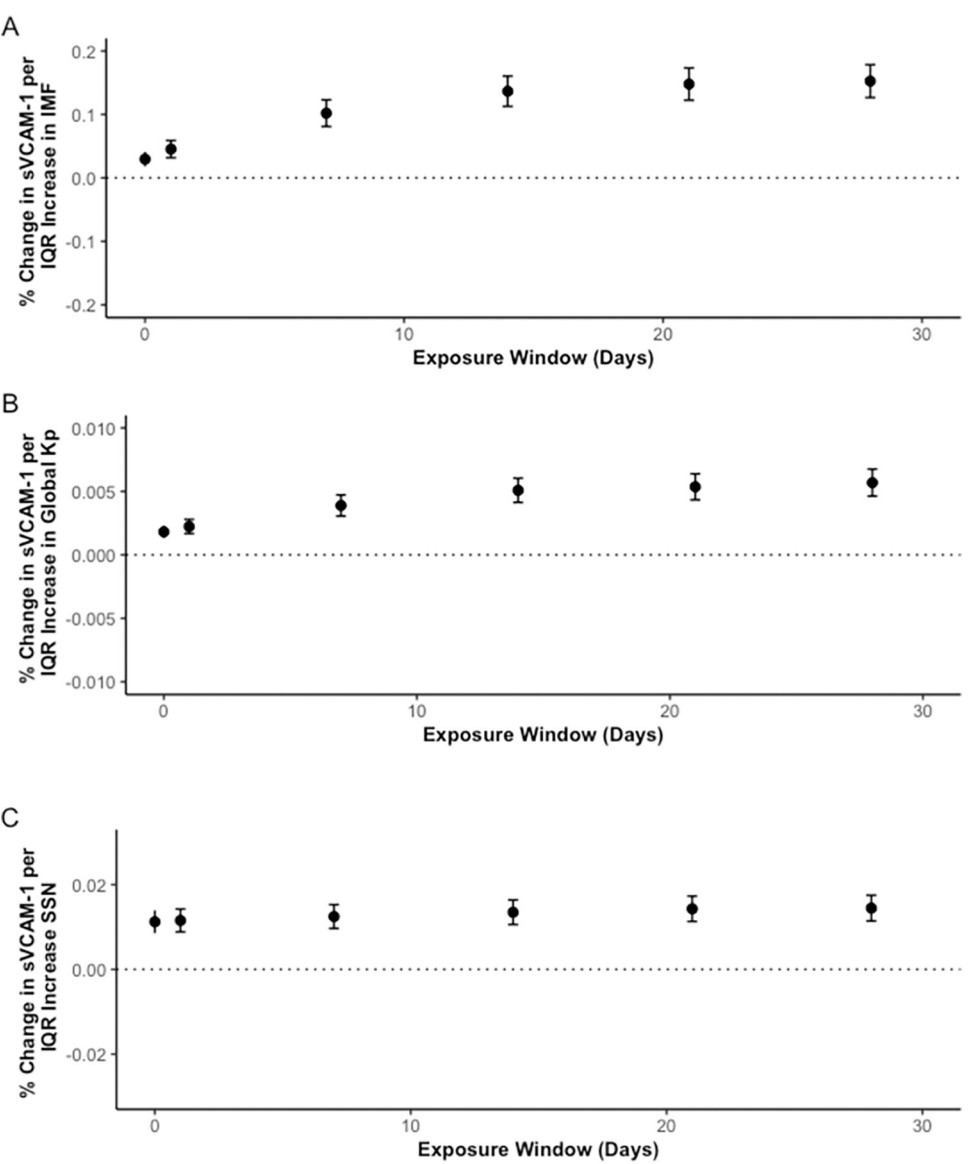

**Fig 2.** Percent change in sVCAM-1 concentrations per IQR increase in IMF (A), $K_p$ index (B), and SSN (C) over 28 day moving averages. * y-axes have different scales.

## C-reactive protein (CRP)

The effects of solar activity exposures on CRP were positive but of lesser magnitude. However, a significant effect was seen for SSN (Fig 3C). On the 28-day moving average on increase an IQR increase in SSN was associated with a 1.34% (0.493, 2.19) change of CRP concentration. IQR increases in IMF overall showed a positive but nonsignificant trend with CRP over the 28 moving day averages (Fig 3A). An IQR increase in $K_p$ index were not associated with a significant percent change of CRP concentration (Fig 3B). Results were of similar magnitude and direction when air pollutants were included in the model. To see all results, refer to S5 Table.

## Fibrinogen

The effects of solar and geomagnetic activity exposures on fibrinogen were not consistent across various solar activity exposure measures. For the 28-day moving average, an IQR increase in $K_p$

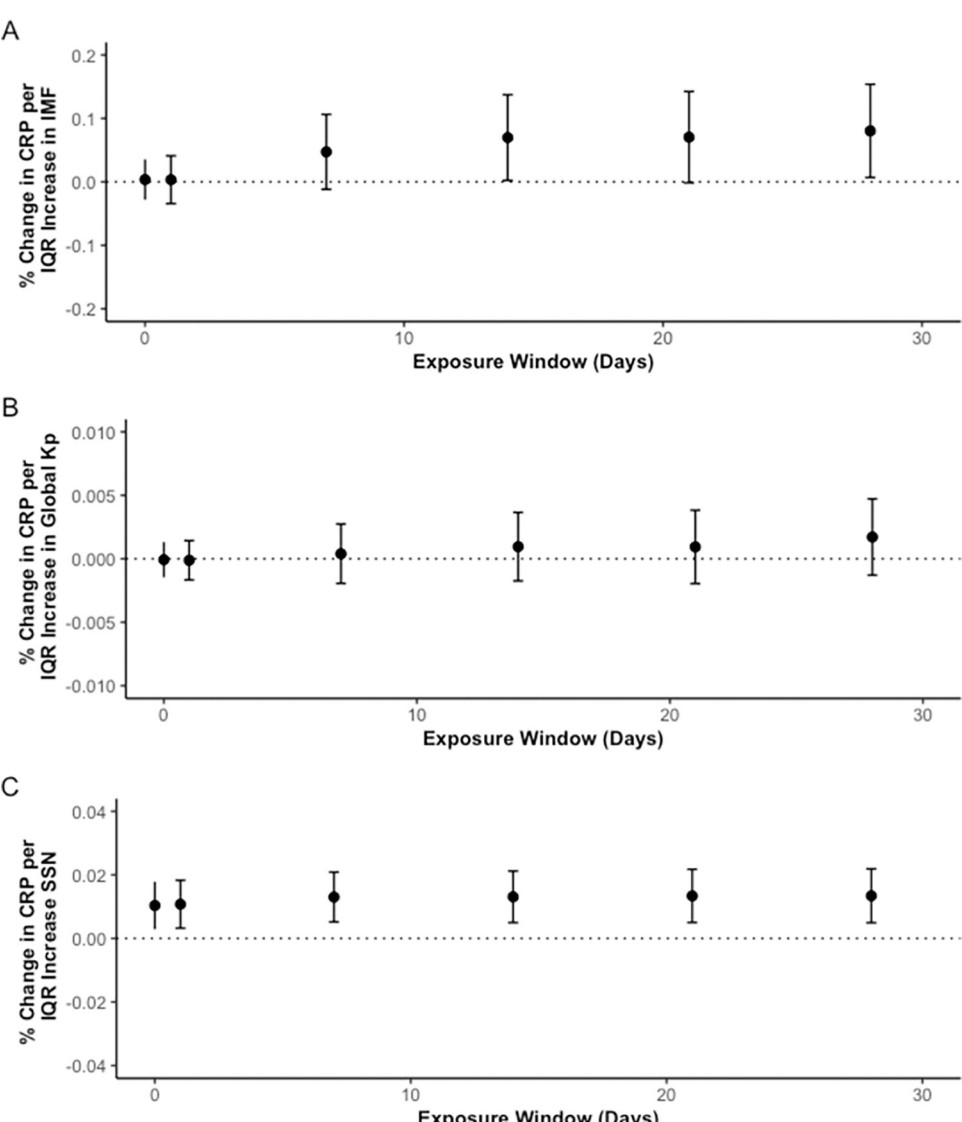

**Fig 3.** Percent change in CRP concentrations per IQR increase in IMF (A), $K_P$ index (B), and SSN (C) over 28 day moving averages. * y-axes have different scales.

was associated with a -0.10% (-0.18, -0.02) change of fibrinogen concentration (Fig 4B). IQR increases in $K_P$ were significantly associated with an overall decrease in fibrinogen concentration when adjusting for particle number. No significant percent change in fibrinogen concentration was associated with an IQR increase in IMF or SSN, however, there was a negative trend seen with IMF and a positive trend seen with SSN (Fig 4C). IQR increases in IMF were not associated with a significant increase of fibrinogen concentrations but had a downward trend over moving day averages (Fig 4A). Results were of similar magnitude and direction when $PM_{2.5}$, BC, PN, and log β were included in the model. To see all results, refer to S6 Table.

## Discussion

To the best of our knowledge, this is the first study to specifically investigate the effects of solar and geomagnetic activity on endothelial activation and inflammatory markers in a large

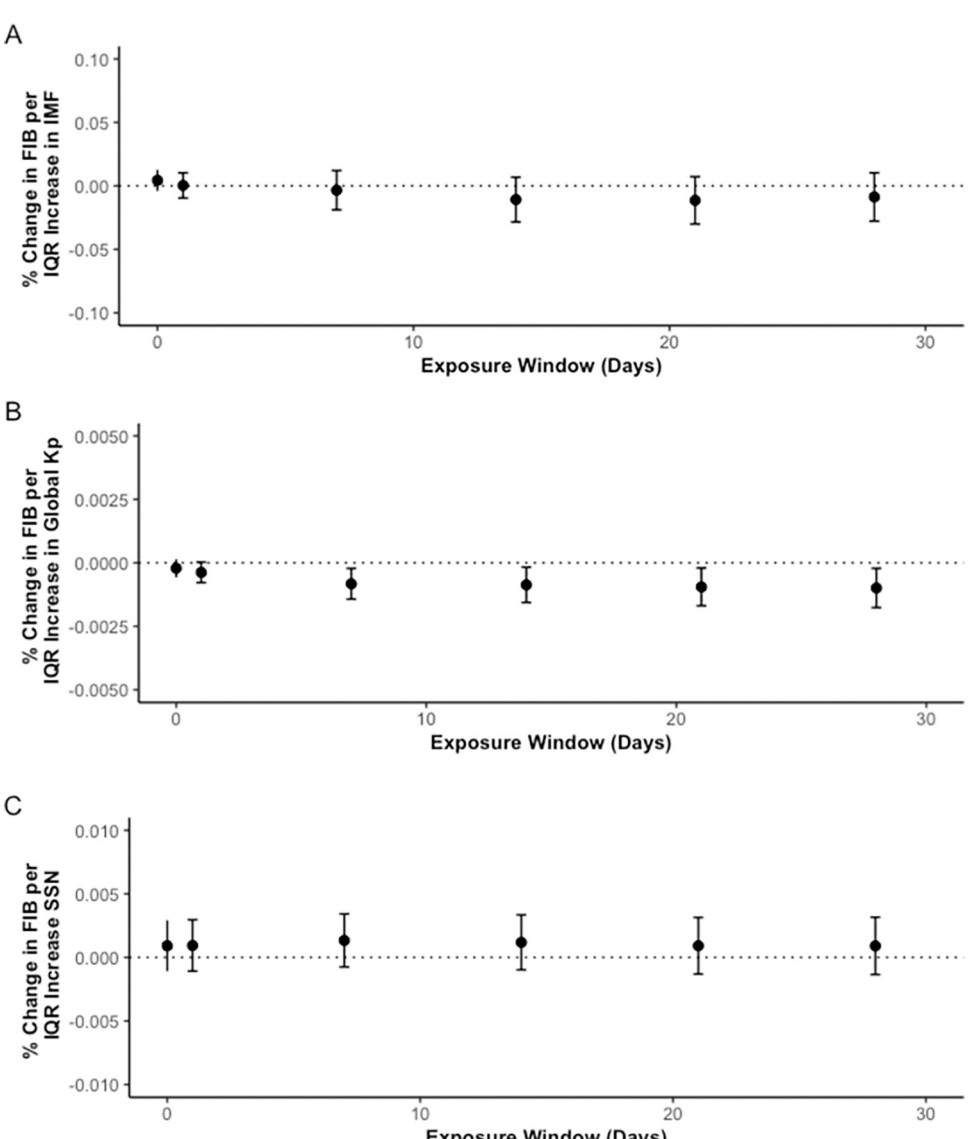

**Fig 4.** Percent change fibrinogen (FIB) concentrations per IQR increase in IMF (A), $K_p$ index (B), and SSN (C) over 28 day moving averages. * y-axes have different scales.

cohort. In this study, we observed positive, significant associations between sICAM-1 and sVCAM-1 concentrations and solar and geomagnetic activity parameters including IMF, SSN, and $K_p$. These associations maintained positive trends and significance in subsequent models adjusting for $PM_{2.5}$, BC, PN, and particle gross β-activity. Associations between solar activity and CRP and fibrinogen were not as consistent nor as strong as the associations seen with sICAM-1 and sVCAM-1. A positive, significant association was observed for CRP and SSN and a positive, but non-significant trend was observed for CRP and $K_p$ index. The association between CRP and IMF was positive and non-significant for shorter moving day averages but had significance with later moving day averages. A significant inverse trend in fibrinogen concentration was observed with exposure to $K_p$. There was an inverse trend seen for fibrinogen and IMF and a positive trend was found for the association between fibrinogen and SSN. The effects of solar activity on endothelial function and inflammatory markers were greater for

later moving day averages suggesting that the impact of solar activity on endothelial and inflammatory markers are delayed. This is similar to those of lagged exposure of air pollutants on inflammatory and vascular markers [45]. This indicates that the effects of solar activity on health outcomes may be delayed by days.

Increases in sICAM-1, sVCAM-1, and CRP concentrations indicate endothelial activation. Although literature in this field is limited, a prior study also showed that CRP levels were significantly correlated with daily geomagnetic activity intensity but not correlated with monthly geomagnetic activity intensity [46]. Other studies have illustrated that physiological and biological systems can be influenced by solar cycles through circadian rhythm disruption, autonomic nervous system signaling, and melatonin production, which impact endothelial function and inflammation [47–49]. A possible explanation for the effects of solar activity on endothelial and inflammatory biomarkers may be through similar biological mechanisms. The autonomic nervous system and melatonin have been found to be sensitive to variations in the geomagnetic activity [50, 51]. Daily autonomic nervous system activity has been shown to respond to changes in solar and geomagnetic activity and to be synchronized with time-varying magnetic fields [48]. Geomagnetic disturbances have also been linked to diurnal variation in melatonin production and decreased melatonin metabolite excretion [52]. Therefore, we theorize that solar activity and geomagnetic disturbances impact the expression of clock genes in suprachiasmatic nuclei (SCN) of the hypothalamus, disrupting the circadian rhythm and melatonin production, contributing to endothelial dysfunction [53, 54].

Melatonin is an antioxidant and an anti-inflammatory endocrine molecule; it modulates reactive oxygen species, reduces oxidative stress, and is sensitive to ionizing radiation in humans [50, 51, 55, 56]. Pineal melatonin secretion is controlled by SCN, which transforms photo cues (i.e. light/dark signals) into melatonin that circulates throughout the body in blood and cerebrospinal fluid, synchronizing peripheral clocks [53]. The endocrine molecule is also part of a feedback loop that helps to maintain SCN function [57]. Vascular tissues have a strong presence of peripheral clocks and; therefore, changes in melatonin production and secretion from dysregulation of the SCN likely have an effect on circadian rhythms, impacting heart and arterial tissue and development of atherosclerosis [58, 59]. Circadian variation has been seen in cell adhesion molecules on epithelial cells, suggesting that these molecules are sensitive to changes in circadian rhythm regulated by the SNC [60].

The link between melatonin, circadian cycles, and cardiac outcomes is hypothesized to be through endothelial dysfunction modulated via oxidative stress as vascular function and the autonomic nervous system are sensitive to oxidative stress [61]. Animal studies have demonstrated that increased oxidative stress reduces vascular function through endothelial damage [61]. Studies have further shown that CRP levels are associated with increased reactive oxygen species, fibrinogen is susceptible to oxidative modification from oxidative stress, and $PM_{2.5}$-induced reactive oxygen species may increase expression of ICAM-1 and VCAM-1 on endothelial cells [62–64]. Thus, the results we observed in this study of the effects of solar and geomagnetic parameters on sICAM-1, sVCAM-1, and CRP are proposed to be the result of the increased reactive oxygen species and oxidative stress due to decreased melatonin concentrations.

However, it is important to note that the impact of solar activity on fibrinogen concentrations were not consistent with the other outcomes in this study. There is limited literature on the effects of solar activity on fibrinogen but other environmental exposures, such as air pollution, demonstrate inconsistent or null effects on fibrinogen concentrations [65–69]. Additionally, a study evaluating exposure of post myocardial infarction patients to artificial whole-body ultraviolet radiation demonstrated that there was no significant change in plasma fibrinogen concentration in response to the artificial UV radiation, which increase during intense solar

activity [70]. Therefore, there could be differences, or variations, in the physiologic response of fibrinogen to solar and geomagnetic activity that result in an inverse response compared to sICAM-1, s-VCAM-1, and CRP.

## Effect of particle exposures

We were concerned that particle exposures would be affected by solar activity and impact the association between endothelial function and inflammation as there has been research reporting effects of particle exposures on endothelial function and inflammation. Air pollution has been shown to increase ICAM-1, VCAM-1, CRP and fibrinogen levels and particle β-activity has been associated with an increased concentrations of inflammatory markers [29, 44, 71]. Air pollutants were included in secondary analyses in this study because recent prior research has indicated that there synergistic effects between $PM_{2.5}$ and radon, and we were concerned of similar effects with solar activity [36]. A recent study suggested that solar irradiance and radiation contribute to atmospheric photochemical reactions, impacting particulate formation [72]. SSN has also been correlated with higher air pollution indices in cities with higher air pollution, with a weaker correlation seen in cities with low air pollution levels [73]. Additionally, melatonin has been shown to modulate the effect of air pollution on oxidative stress, suggesting that melatonin can be associated with both oxidative stress and environmental factors [74]. However, our analyses demonstrated that both in the presence and absence of air pollution variables, solar and geomagnetic activity had a significant association with the health outcomes. Our study suggests that solar and geomagnetic activity may also contribute, independently from air pollutants and β radiation, to changes in sICAM-1, sVCAM-1, and CRP levels.

## Public health implications

The public health implications of this study are of great importance. Solar activity exposure is ubiquitous and its effects on endothelial activation and function and inflammatory markers add to our understanding of the causes of cardiac outcomes, especially as cardiovascular diseases are the leading cause of global mortality [75].

We note the CRP is well established as a circulating biomarker of systemic inflammation. In epidemiologic studies, patients with higher levels of CRP, among a continuum, have been linked to cardiovascular events [76, 77]. CRP has also been shown to increase sICAM-1 and sVCAM-1 concentrations in endothelial cells in the presence of serum, suggesting that CRP concentrations affect other inflammatory components and the progression of atherosclerosis [78]. Another study demonstrated that study subjects that had both elevated levels of sICAM-1 and sVCAM-1 had increased risk of CHD after adjusting for other CHD factors compared to subjects who had elevated levels of one marker and not the other, postulating that elevated concentrations of both endothelial adhesion molecules could be associated with progression of CHD [79]. Other studies have indicated that sICAM-1 could be associated with atherosclerosis, CHD, and future cardiac events [76, 80].

The results broaden our understanding of factors that could contribute to the development of cardiovascular disease. The results of this study suggest that clinicians may need to include solar cycle periodicity as a cardiovascular risk factor. This novel study contributes to the current knowledge of how environmental exposures impact endothelial function and inflammation.

## Strengths and limitations

This is the first study to our knowledge to investigate the impact of solar and geomagnetic activity on endothelial cell function and inflammatory markers after adjusting for air pollution

and particle radioactivity. The study was conducted in Massachusetts with little spatial variability in intensity of solar and geomagnetic parameters. Additionally, it was conducted in an established mostly white male cohort with limited ethnic and racial diversity and with no underlying co-morbidities. We only had data from one solar cycle and, ideally, to study solar cycle impact we would need data for at least two solar cycles, the equivalent of 22-years. However, since we wanted to evaluate the impacts of air pollution in our analyses, we were unable to study two full solar cycles as we only had air pollution data starting from year 2000. Furthermore, the solar electromagnetic radiation released takes hours to a few days to arrive to Earth, in contrast to the $K_p$ index, which is related to the Earth's magnetic sphere with perturbations impacting the population directly (i.e. within minutes to a few hours).

### Future directions

Since the population was 98% white and all male, additional studies need to be conducted in diverse populations (women, children, ethnic minorities, etc.) to better understand the impacts of solar activity on health outcomes. Moreover, this study was conducted in a small region and does not represent exposures in different latitudes. Since solar and geomagnetic activity-related electromagnetic radiation varies based on latitude and altitude, further studies are necessary to understand these associations at different geospatial scales.

### Conclusion

This study has demonstrated that solar activity and geomagnetic disturbances significantly increase sICAM-1 and sVCAM-1 levels. Solar activity was also associated with a significant increase in CRP and geomagnetic disturbance ($K_p$ index) was associated with a significant increase in fibrinogen. These findings may help explain temporal increase of inflammatory and endothelial markers during intense solar and geomagnetic activity.

### Supporting information

**S1 Table. Most significant moving average window for air pollutants and log β, 2000–2017.**
(DOCX)

**S2 Table. Distribution of solar, particulate air pollution, and β activity variables, measured between 2000 and 2017.**
(DOCX)

**S3 Table. Percent change (estimate\*IQR\*100) of sICAM-1 associated per IQR increase (95% CI) of exposure variable.**
(DOCX)

**S4 Table. Percent change (estimate\*IQR\*100) of sVCAM-1 associated per IQR increase (95% CI) of exposure variable.**
(DOCX)

**S5 Table. Percent change (estimate\*IQR\*100) of CRP associated per IQR increase (95% CI) of exposure variable.**
(DOCX)

**S6 Table. Percent change (estimate\*IQR\*100) of fibrinogen associated per IQR increase (95% CI) of exposure variable.**
(DOCX)

## Acknowledgments

The VA Normative Aging Study is sponsored by the U.S. Department of Veterans Affairs Cooperative Studies Program/ERIC and is as research component of the Massachusetts Veterans Affairs Epidemiology Research Center (MAVERIC).

## Author Contributions

**Conceptualization:** Jessica E. Schiff, Carolina L. Z. Vieira, Petros Koutrakis.

**Data curation:** Carolina L. Z. Vieira, Pantel Vokonas, Petros Koutrakis.

**Formal analysis:** Jessica E. Schiff, Carolina L. Z. Vieira, Eric Garshick, Veronica Wang, Samantha M. Tracy, Petros Koutrakis.

**Funding acquisition:** Petros Koutrakis.

**Methodology:** Jessica E. Schiff, Carolina L. Z. Vieira, Veronica Wang, Petros Koutrakis.

**Software:** Jessica E. Schiff, Veronica Wang.

**Supervision:** Carolina L. Z. Vieira, Petros Koutrakis.

**Validation:** Carolina L. Z. Vieira.

**Writing – original draft:** Jessica E. Schiff, Carolina L. Z. Vieira, Petros Koutrakis.

**Writing – review & editing:** Jessica E. Schiff, Carolina L. Z. Vieira, Eric Garshick, Veronica Wang, Annelise Blomberg, Diane R. Gold, Joel Schwartz, Samantha M. Tracy, Pantel Vokonas, Petros Koutrakis.

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
