## [Decision Letter · Decision Letter 0]

9 Dec 2021

PONE-D-21-14987The Role of Solar Activity in Endothelial Activation and Inflammation in the NAS CohortPLOS ONE

Dear Dr. Schiff,

Thank you for submitting your manuscript to PLOS ONE. After careful consideration, we feel that it has merit but does not fully meet PLOS ONE’s publication criteria as it currently stands. Therefore, we invite you to submit a revised version of the manuscript that addresses the points raised during the review process.

We look forward to receiving your revised manuscript.

Kind regards,

Andreas Zirlik, MD

Academic Editor

PLOS ONE

Journal Requirements:

Reviewers' comments:

Reviewer's Responses to Questions

**Comments to the Author**

1. Is the manuscript technically sound, and do the data support the conclusions?

Reviewer #1: Yes

Reviewer #2: Partly

2. Has the statistical analysis been performed appropriately and rigorously? 

Reviewer #1: Yes

Reviewer #2: Yes

3. Have the authors made all data underlying the findings in their manuscript fully available?

Reviewer #1: Yes

Reviewer #2: No

4. Is the manuscript presented in an intelligible fashion and written in standard English?

Reviewer #1: Yes

Reviewer #2: Yes

5. Review Comments to the Author

Reviewer #1: This is a very interesting manuscript pointing to environmental challenges widely underrecognized. The broad data are well presented and the association of inflammatory markers and solar activity appears intriguing.

However, there are a few points I would like to raise.

Why was the adjustment of weather variables limited to temperature and humidity. Did the authors have access to other parameters such as air pressure and were these analysed, too?

Similarly, were the pollution recordings limited to PM2.5, black carbon and PN or did the authors have access and/or analysed other parameters like ozone, SO2, NO or others that might also impact with inflammation.

Where did the participants of the study live. Impact of solar activity can be considered very similar in a wide region, whereas temperature and humidity can already vary significantly within smaller distances and air pollution must be considered to be very heterogenous even within small distances. Therefore, air pollution parameters measured at one certain localisation might not well represent patients` exposure living in a e.g. more rural or distant location.

Taking the previously published Enviromental Research manuscript of the study group into consideration it seems likely that a plethora of parameters was analysed. Was there a prospective statistical plan for the current analysis focussing on the parameters presented? And has that been limited to the presented parameters? Or how many parameters had been analysed to find the association described in the manuscript. This seems to be important to rule out chance findings.

Moreover, it is stated in the discussion that the manuscript is the first study to investigate the effects of solar activity on … inflammatory markers in a large cohort despite the fact that the same group recently released a very similar manuscript (Enviromental Research) based on the same cohort and measurements describing significant changes in white blood cells which can be counted as part of the inflammatory system. This should be clearly stated. And I recommend to discuss if there might be a direct link between changes in white blood parameters and the inflammatory markers presented in this manuscript.

Reviewer #2: I have reviewed the manuscript "The Role of Solar Activity in Endothelial Activation and Inflammation in the NAS Cohort" by Schiff et al.

In this manuscript, the authors shed some light on the associations between solar activity exposures and endothelial and inflammatory biomarkers in the Normative Aging Study (NAS) cohort. The paper addresses the hitherto undervalued relationship between solar activity and inflammation, whereas the latter plays a crucial role in developing atherosclerosis and its downstream consequences.

A key finding of this study is that solar activity was associated with significantly increased levels of ICAM-1s, VCAM-1s, and CRP. Further, geomagnetic disturbance associates with an increase of fibrinogen in the study cohort examined. In a second analysis, the authors considered the influence of air pollution, thus excluding any possible impact on their previous results.

In principle, this study is a beautiful piece of work at the interplay of environmental exposures and inflammation/vascular activity and, therefore, another part of the puzzle to understand the causes of atherosclerosis. There are, however, several shortcomings in the discussion regarding the functional consequences and the translational relevance of the findings. In addition, it is strongly recommended to revise the manuscript's structure in terms of figures, abbreviations, and tables.

Major concerns

The authors rightfully state that understanding the exposures that contribute to inflammation and the development of atherosclerosis is of interest regarding the diagnosis and treatment of cardiovascular disease. The present study examines the impact of solar activity and geomagnetic disturbances on inflammation and endothelial activation. As the authors point out, these environmental exposures are ubiquitous and associated with a significant increase of ICAM-1s, VCAM-1s, and CRP in the present study. What are the consequences of these findings? How can these results help to improve the prevention and monitoring of cardiovascular disease, as exposure is ubiquitous? Can these results be translated into a clinical context? The authors are kindly asked to discuss these points in detail.

The authors speculate about the causal mechanism behind the associations of increased biomarkers and solar activity in the discussion. They convincingly argue that solar activity could lead to downstream impacts via disruption of the circadian rhythm. Such explanations are welcome if they are substantiated by other works, as in the present manuscript. However, hypotheses about mechanisms should be removed from the conclusion since the authors do not provide any evidence from their data.

Minor concerns

In the method section "Blood Measurements", the authors do not provide information on the clinical context of the blood collection. Furthermore, the information on which days the blood samples were taken is missing.

In Figures 1 and 2, the capital letters A & B are depicted on top of the associated plots. However, no reference is made to the letters in the text or the legend. Figures 3 and 4 also show letters that are only partially referred to in the text. Please refer to the letters of your figures both in the text and in the legend. The correct reference would facilitate reading flow and improve the clarity of the figures and corresponding text passages.

I understand why the authors listed all their results considering air pollution (see. Table 2-6). This data is of interest, but the connection to the principal objective of this manuscript (Solar & Geomagnetic activity and Inflammation & Endothelial Activation) is not fully evident. In this reviewer's opinion, the tables are more suitable as supplemental figures.

Please change ICAM and VCAM in Figures 1 & 2 to sICAM-1 and sVCAM-1. The authors are asked to use consistent abbreviations in the present manuscript. There are four different abbreviations used in the title, text, and figure (e.g., VCAM-1, sVCAM-1, sVCAM, VCAM)

Line 36, page 2: "These results demonstrate that solar activity might be upregulating endothelial activation and inflammation through altering, or disrupting, circadian rhythms." The second half of the sentence is pure speculation without any valid evidence provided by the authors in their manuscript.

Page 3, line 92: "…bind white blood cells and lymph cells…" Please change lymph cells to lymphocytes.

Page 10, line 221: “mean sICAM-1, sVCAM-1, CRP, and fibrinogen concentration was 311.5 ng/L…” The structure of this sentence is not intuitive. Please rephrase. It is sufficient to mention only the most interesting baseline characteristics and refer to the descriptive statistics in table 1.

Page 12, line 263: "…QR increase in SSN…" vs. Page 12, line 264: "…IQR increase…". Please use consistent abbreviations

Page 27, line 527: "…a future study could include evaluate…" Unclear. Please rephrase and change the wording.

6. PLOS authors have the option to publish the peer review history of their article (what does this mean?). If published, this will include your full peer review and any attached files.

Reviewer #1: No

Reviewer #2: No

---

## [Author Response · Author response to Decision Letter 0]

30 Apr 2022

Response to Reviewer Comments:

I would like to thank the reviewers for taking time to review our submitted manuscript “The Role of Solar Activity in Endothelial Activation and Inflammation in the NAS Cohort”. We especially appreciate the many constructive and insightful comments, suggestions and corrections from the reviewers. The revised manuscript is significantly improved by their input.

Please refer to the document titled “Revised Manuscript with Track Changes” when reviewing response to reviewer comments to see all changes. 

Responses Initial Comments: 

1. Have the authors made all data underlying the findings in their manuscript fully available? The PLOS Data policy requires authors to make all data underlying the findings described in their manuscript fully available without restriction, with rare exception (please refer to the Data Availability Statement in the manuscript PDF file). The data should be provided as part of the manuscript or its supporting information, or deposited to a public repository. For example, in addition to summary statistics, the data points behind means, medians and variance measures should be available. If there are restrictions on publicly sharing data—e.g. participant privacy or use of data from a third party—those must be specified.

Reviewer #1: Yes

Reviewer #2: No

Response: Human subject data collected by the US Department of Veterans Affairs (VA) studies cannot be publicly shared as result of VA policies. However, deidentified data may be shared with VA-credentialled researchers under a data sharing agreement if approved by an institutional review board 

Changes to Reference List: 

Several references were added to the manuscript in the process of revisions. Added references include citations: 13, 28, 40, 78, 79, 80. 

Responses to Reviewer 1: 

1. Why was the adjustment of weather variables limited to temperature and humidity. Did the authors have access to other parameters such as air pressure and were these analysed, too?

Response: The adjustment of weather variables was limited to temperature and humidity because those variables are the most common parameters included in other similar studies for environmental factors, such as air pollution exposures and solar activity. The authors also did not have access to other meteorological data such as air pressure. 

The added text is included below (page 8 line 178-181): 

“Adjustment of weather variables was limited to temperature and humidity because those variables are the most common meteorological parameters used in similar studies evaluating the impacts of solar and geomagnetic activity on health outcomes(5,38,39). Furthermore, the authors did not have access to other parameters, such as air pressure.”

2. Similarly, were the pollution recordings limited to PM2.5, black carbon and PN or did the authors have access and/or analysed other parameters like ozone, SO2, NO or others that might also impact with inflammation.

Response: The authors limited the pollution analysis to particulate matter because prior research from the same group suggests that solar activity may interact with particulate matter. Therefore, we wanted to adjust for the pollutants in the analysis to evaluate the impact of solar geomagnetic activity controlling for pollution as it is already known that pollution has known health impacts. The researchers were also not provided with this data. 

The added text is included below (page 7 line 153-156): 

“We specifically limited the pollution analysis to particulate matter constituents as prior research suggests that solar activity may interact with particulate matter, and thus we adjusted for these pollutants in our analysis(35).”

3. Where did the participants of the study live. Impact of solar activity can be considered very similar in a wide region, whereas temperature and humidity can already vary significantly within smaller distances and air pollution must be considered to be very heterogeneous even within small distances. Therefore, air pollution parameters measured at one certain localisation might not well represent patients` exposure living in a e.g. more rural or distant location.

Response: Although participants lived in Massachusetts upon enrollment into the NAS cohort, participants moved throughout the study period (see pages 8-9 lines 214-217 for description in the text). The participants lived in geographically different regions of the US, however, a sensitivity analysis was conducted to evaluate if moving from Massachusetts impacted the results (see pages 8-9 lines 214-217 for description in the text). Some participants moved from Massachusetts to other states after the cohort was established, although most returned to Massachusetts for the follow up visits. The sensitivity analysis of location of residence revealed slightly larger effect estimates when restricted to only those residing in Massachusetts, however, the number of observations in the study decreased to 1,653, reducing our power (see page 10 lines 229-231). We elected to include all participants because the changes in effect estimates were minimal and to preserve power.

4. Taking the previously published Environmental Research manuscript of the study group into consideration it seems likely that a plethora of parameters was analysed. Was there a prospective statistical plan for the current analysis focussing on the parameters presented? And has that been limited to the presented parameters? Or how many parameters had been analysed to find the association described in the manuscript. This seems to be important to rule out chance findings.

Response: (Please note that I am a co-author on the Environmental Research manuscript). Yes, while there were many parameters analyzed in this study, there was a prospective statistical plan for the current analysis focusing on the parameters presented. This individual study was part of a larger group of studies that are evaluating the impacts of solar geomagnetic activity on various health outcomes including immune function, blood pressure, and cardiac outcomes. In particular this study evaluated markers for endothelial activation and inflammation, ICAM-1, VCAM-1, CRP, and Fibrinogen, independently of immune markers. Initially, IL-6 was also evaluated in addition to the four markers previously mentioned, however, it was challenging to manage and present data for five markers, and the results were not as clear for IL-6, so it was dropped. The other parameters included in the study, including the main exposures and covariates, remained the same throughout the analysis, and are similar across this study and the immune function study published in the Environmental Research Manuscript. There are some differences in covariates between this study and the immune function and that is due to needing to adjust for specific covariates that are pertinent to immune function which are not as applicable to endothelial activation or inflammation, such as blood cancers. 

5. Moreover, it is stated in the discussion that the manuscript is the first study to investigate the effects of solar activity on … inflammatory markers in a large cohort despite the fact that the same group recently released a very similar manuscript (Enviromental Research) based on the same cohort and measurements describing significant changes in white blood cells which can be counted as part of the inflammatory system. This should be clearly stated. And I recommend to discuss if there might be a direct link between changes in white blood parameters and the inflammatory markers presented in this manuscript.

Response: Yes, this is an important observation. This manuscript was submitted prior to the acceptance and publication of the Environmental Research manuscript and thus, this statement needs to be edited and updated to reflect the new information. I will clarify that this study is specifically looking at markers for endothelial activation and inflammation. 

The added text is included below (pages 3-4 lines 69-72): 

“This study complements recent research evaluating the associations between solar and geomagnetic activity and immune function and is the first study to our knowledge that specifically investigates the associations between solar activity and endothelial and inflammation (13).”

We did not explore direct links between changes in the white blood parameters and the inflammatory markers presented in this manuscript. However, it is important to acknowledge that the two are physiologically closely related and that changes in the white blood parameters could impact changes in the endothelial parameters and vice versa, and that this could be an avenue for future research.

The added text is included below (page 4 lines 72-75): 

“While inflammation and immune response are physiologically closely related and that changes in white blood parameters could impact inflammatory and endothelial parameters and vice versa, this study focuses only on the impacts of solar and geomagnetic activity on endothelial parameters.” 

Responses to Reviewer 2: 

1. The authors rightfully state that understanding the exposures that contribute to inflammation and the development of atherosclerosis is of interest regarding the diagnosis and treatment of cardiovascular disease. The present study examines the impact of solar activity and geomagnetic disturbances on inflammation and endothelial activation. As the authors point out, these environmental exposures are ubiquitous and associated with a significant increase of ICAM-1s, VCAM-1s, and CRP in the present study. What are the consequences of these findings? How can these results help to improve the prevention and monitoring of cardiovascular disease, as exposure is ubiquitous? Can these results be translated into a clinical context? The authors are kindly asked to discuss these points in detail.

Response: There are health consequences related to increased concentrations of sICAM-1 and sVCAM-1, with research showing elevated levels of sICAM-1 associated with increased risk of cardiovascular events and increased concentrations of both sICAM-1 and sVCAM-1 increased risk of CHD. Elevated levels of CRP have also been associated with increased risk of CHD and cardiovascular events. 

The added text is included below (pages 21-22 lines 449-459):

“We note the CRP is well established as a circulating biomarker of systemic inflammation. In epidemiologic studies in patients with CRP, higher levels of CRP among a continuum have been linked to cardiovascular events (76,77). CRP has also been shown to increase sICAM-1 and sVCAM-1 concentrations in endothelial cells in the presence of serum, suggesting that CRP concentrations affect other inflammatory components and the progression of atherosclerosis (78). Shai et. al. (2006) demonstrated that study subjects that had both elevated levels of sICAM-1 and sVCAM-1 had increased risk of CHD after adjusting for other CHD factors compared to subjects who had elevated levels of one marker and not the other, postulating that elevated concentrations of both endothelial adhesion molecules could be associated with progression of CHD (79). Other studies have indicated that sICAM-1 could be associated with atherosclerosis, CHD, and future cardiac events (76,80).”

How can these results help to improve the prevention and monitoring of cardiovascular disease, as exposure is ubiquitous? Can these results be translated into a clinical context? 

The results found by this study can improve the monitoring of cardiovascular disease by broadening our understanding of all the factors that could contribute to the development of cardiovascular disease. This study suggests that natural exposures to geomagnetic activity are associated with increased concentrations of endothelial adhesion molecules and inflammatory markers that have previously been shown to increase risk for cardiovascular events and CHD. Although the exposure in this study is ubiquitous and is not a lifestyle factor that can be adjusted for, it provides an understanding of backgrounds factors that may elevate CVD risk that cannot currently be managed and allows researchers to better understand the synergistic effects of different exposure and lifestyle factors on the development of CVD. 

The added text is included below (pages 22 lines 461-464):

“The results broaden our understanding of factors that could contribute to the development of cardiovascular disease. The results of this study suggest that clinicians may need to include solar cycle time period as a cardiovascular risk factor. This novel study contributes to the current knowledge of how environmental exposures impact endothelial function and inflammation.”

2. The authors speculate about the causal mechanism behind the associations of increased biomarkers and solar activity in the discussion. They convincingly argue that solar activity could lead to downstream impacts via disruption of the circadian rhythm. Such explanations are welcome if they are substantiated by other works, as in the present manuscript. However, hypotheses about mechanisms should be removed from the conclusion since the authors do not provide any evidence from their data.

The text in the conclusion was edited to reflect the comment above, removing hypotheses about mechanisms. 

The edited text is included below (page 23 lines 491-495):

“This study has demonstrated that solar activity and geomagnetic disturbances significantly increase sICAM-1 and sVCAM-1 levels. Solar activity was also associated with a significant increase in CRP and geomagnetic disturbance (Kp index) was associated with a significant increase in fibrinogen. These findings may help explain temporal increase of inflammatory and endothelial markers during intense solar and geomagnetic activity.”

Minor concerns from Reviewer 2

1. In the method section "Blood Measurements", the authors do not provide information on the clinical context of the blood collection. Furthermore, the information on which days the blood samples were taken is missing.

All participants were free of chronic conditions at baseline and attended clinical physical examinations every 3–5 years. The dataset used for the analysis contains the date of visit for each study participant and the date of visit is linked to the date that blood samples were collected. The clinical physical examinations were conducted every 3-5 years, with the blood samples collected at each clinical visit from participants after an overnight fasting period. 

The edited text is included below (page 4 lines 81-90):

“At recruitment, men were free of chronic disease at baseline, had a mean age of 42 (range 21-81), and had subsequent examinations every 3 to 5 years. The participants abstained from smoking, were asked to eat a fat-free meal the night before, and completed an overnight fast prior to their clinical examination (16). Examinations occurred on Tuesdays and Wednesdays and the dates that the bloods were collected were matched to the dates of the solar activity, air pollution, and weather measurements. A complete physical exam and laboratory testing, including phlebotomy, were conducted at each clinical visit. Standardized questionnaires were administered to each participant to collect data about medical history, smoking history, and alcohol consumption. Only participants with recorded observations between May 2000 and December 2017 were included in the study.” 

1. In Figures 1 and 2, the capital letters A & B are depicted on top of the associated plots. However, no reference is made to the letters in the text or the legend. Figures 3 and 4 also show letters that are only partially referred to in the text. Please refer to the letters of your figures both in the text and in the legend. The correct reference would facilitate reading flow and improve the clarity of the figures and corresponding text passages.

Please see the manuscript with track changes to see the improved labeling of figures. 

2. I understand why the authors listed all their results considering air pollution (see. Table 2-6). This data is of interest, but the connection to the principal objective of this manuscript (Solar & Geomagnetic activity and Inflammation & Endothelial Activation) is not fully evident. In this reviewer's opinion, the tables are more suitable as supplemental figures.

Tables 2-6 were removed and added as supplementary figures. 

3. Please change ICAM and VCAM in Figures 1 & 2 to sICAM-1 and sVCAM-1. The authors are asked to use consistent abbreviations in the present manuscript. There are four different abbreviations used in the title, text, and figure (e.g., VCAM-1, sVCAM-1, sVCAM, VCAM)

Thank you for the note. The manuscript was edited for consistent abbreviations where appropriate. A few locations were left as ICAM-1 or VCAM-1 as that is how the markers were reported in the cited literature. Please refer to the revised manuscript to see the changes in abbreviations. 

4. Line 36, page 2: "These results demonstrate that solar activity might be upregulating endothelial activation and inflammation through altering, or disrupting, circadian rhythms." The second half of the sentence is pure speculation without any valid evidence provided by the authors in their manuscript.

Please see the edits included below (page 2, lines 36-37):

“These results demonstrate that solar activity might be upregulating endothelial activation and inflammation.”

5. Page 3, line 92: "…bind white blood cells and lymph cells…" Please change lymph cells to lymphocytes.

The added text is included below (page 4 lines 102-104):

“They are surface receptors on endothelial cells that bind white blood cells and lymphocytes to the endothelium, generating inflammation and plaque buildup (18–22).”

6. Page 10, line 221: “mean sICAM-1, sVCAM-1, CRP, and fibrinogen concentration was 311.5 ng/L…” The structure of this sentence is not intuitive. Please rephrase. It is sufficient to mention only the most interesting baseline characteristics and refer to the descriptive statistics in table 1.

The added text is included below (page 11 lines 239-241):

“At first visit, mean sICAM-1 concentration was 311.5 (SD=80.8) ng/L , mean s-VCAM-1 was 1,085.1 (SD=377.8) ng/L, mean CRP was 2.3 (SD=2.0) mg/L, and mean fibrinogen 342.7 (SD=78.2) mg/dL.”

7. Page 12, line 263: "…QR increase in SSN…" vs. Page 12, line 264: "…IQR increase…". Please use consistent abbreviations

The added text is included below (page 13 lines 260-261):

“Overall, IQR increases in SSN, IMF, and Kp were associated with a significant, positive percent change in sICAM-1 concentration (Figure 1.A-C).”

8. Page 27, line 527: "…a future study could include evaluate…" Unclear. Please rephrase and change the wording.

Edited future directions text (page 23 lines 483-488):

“Since the population was 98% white and all male, additional studies need to be conducted in diverse populations (women, children, ethnic minorities, etc.) to better understand the impacts of solar activity on health outcomes. Moreover, this study was conducted in a small region and does not represent exposures in different latitudes. Since solar and geomagnetic activity-related electromagnetic radiation varies based on latitude and altitude, further studies are necessary to understand these associations at different geospatial scales.”

---

## [Editor Report · Decision Letter 1]

6 May 2022

The Role of Solar Activity in Endothelial Activation and Inflammation in the NAS Cohort

PONE-D-21-14987R1

Dear Dr. Schiff,

We’re pleased to inform you that your manuscript has been judged scientifically suitable for publication and will be formally accepted for publication once it meets all outstanding technical requirements.

Kind regards,

Andreas Zirlik, MD

Academic Editor

PLOS ONE
---

## [Editor Report · Acceptance letter]

15 Jul 2022

PONE-D-21-14987R1 

The Role of Solar and Geomagnetic Activity in Endothelial Activation and Inflammation in the NAS Cohort 

Dear Dr. Schiff:

I'm pleased to inform you that your manuscript has been deemed suitable for publication in PLOS ONE. Congratulations! Your manuscript is now with our production department. 

Kind regards, 

on behalf of

Univ. Prof. Dr. Andreas Zirlik 

Academic Editor

PLOS ONE